# Particulate matter 2.5, metropolitan status, and heart failure outcomes in US counties: A nationwide ecologic analysis

**Edward W. Chen**[ID]<sup></sup>, **Khansa Ahmad**[ID]<sup></sup>, **Sebhat Erqou**, **Wen-Chih Wu**[ID]*

The Providence Veterans Affairs Medical Center, Lifespan Hospitals and the Warren Alpert Medical School at Brown University, Providence, Rhode Island

☯ These authors contributed equally to this work.
* wen-chih.wu@va.gov

**Data Availability Statement:** All data are publicly available from the CDC and United States Census Bureau (https://nccd.cdc.gov/DHDSPAtlas/?state=County&ol=%5b10).

## Abstract

The relationship between particulate matter with a diameter of 2.5 micrometers or less (PM$_{2.5}$) and heart failure (HF) hospitalizations and mortality in the US is unclear. Prior studies are limited to studying the effects of daily PM$_{2.5}$ exposure on HF hospitalizations in specific geographic regions. Because PM$_{2.5}$ can vary by geography, this study examines the effects of annual ambient PM$_{2.5}$ exposure on HF hospitalizations and mortality at a county-level across the US. A cross-sectional analysis of county-level ambient PM$_{2.5}$ concentration, HF hospitalizations, and HF mortality across 3135 US counties nationwide was performed, adjusting for county-level demographics, socioeconomic factors, comorbidities, and healthcare-associated behaviors. There was a moderate correlation between county PM$_{2.5}$ and HF hospitalization among Medicare beneficiaries (r = 0.41) and a weak correlation between county PM$_{2.5}$ and HF mortality (r = 0.08) (p-values < 0.01). After adjustment for various county level covariates, every 1 ug/m$^3$ increase in annual PM$_{2.5}$ concentration was associated with an increase of 0.51 HF Hospitalizations/1,000 Medicare Beneficiaries and 0.74 HF deaths/100,000 residents (p-values < 0.05). In addition, the relationship between PM$_{2.5}$ and HF hospitalizations was similar when factoring in metropolitan status of the counties. In conclusion, increased ambient PM$_{2.5}$ concentration level was associated with increased incidence of HF hospitalizations and mortality at the county level across the US. This calls for future studies exploring policies that reduce ambient particulate matter pollution and their downstream effects on potentially improving HF outcomes.

## Introduction

Particulate matter pollutant is defined as a mixture of liquid and solid particles including smoke, dirt, and microscopic chemicals that reside in the air [1]. These are generated from many sources, including directly from unpaved roads, fires, and construction sites as well as an end product of chemical reactions involving emissions from automobiles, power plants, and factories [1]. Particulate matter has been shown to be associated with various respiratory illness [2]. Particulate matter pollutants with a diameter of 2.5 micrometers or less (PM$_{2.5}$) are especially deleterious to cardiovascular health through many proposed mechanisms, such as

**Funding:** Mr. Edward W Chen received a National Institutes of Health (NIH), T35 (HL094308) grant of $4,747.50. The funders had no role in study design, data collection and analysis, decision to publish, or preparation of the manuscript.

**Competing interests:** The authors have declared that no competing interests exist.

its ability to translocate through the lung alveoli, induce lung inflammation, and activate the sympathetic nervous system, ultimately causing systemic inflammation that leads to acute and chronic effects on the cardiovascular system [3]. The exact relationship between $PM_{2.5}$ and heart failure (HF) specifically, however, has yet to be fully understood and quantified.

Heart failure affects approximately 6.2 million adults in the United States [4] and was noted in 13.4% of all US death certificates in 2018 [4]. The national cost of heart failure was estimated to be $30.7 billion in 2012 [5]. Research based in the United Kingdom has shown a positive association between annual exposure to $PM_{2.5}$ and HF incidence [6]. Although there have been some regional US studies examining the association between daily measures of $PM_{2.5}$ and HF mortality [7–10], there are no studies that have examined nationwide data for the association between long-term exposure to $PM_{2.5}$ and HF hospitalization and mortality. Thus, we undertook an investigation of the association of annual exposure to $PM_{2.5}$ with both HF hospitalizations and HF mortality at a county level nationwide across the US. As it is well-established that major metropolitan areas experience the worst ambient air pollution, which in turn is associated with worse health outcomes [11,12], we also aimed to investigate the interplay of metropolitan status in the association of long-term exposure to $PM_{2.5}$ with HF outcomes.

## Methods

We performed a cross-sectional analysis on data from 3141 US counties from 2010–2015. We excluded US territories and six counties that were missing HF mortality data from this study. All data is publicly accessible via the Centers for Disease Control and Prevention (CDC) [13] and the US Census Bureau [13].

### Exposure of interest

Annual county $PM_{2.5}$ concentration level averages (μg/m$^3$) for 2014 were obtained from the CDC. This data was originally obtained by the Environmental Protection Agency's (EPA) Air Quality System (AQS) monitors, which measure pollutant levels at varying frequencies (every third day, sixth day, or everyday depending on surrounding pollutant concentration) and report concentrations as 24-hour averages. Community Multiscale Air Quality (CMAQ) modeling was used to predict $PM_{2.5}$ levels for counties without monitoring systems. Annual averages were calculated from these 24-hour samples and published by the CDC [13]. Though short-term elevations in $PM_{2.5}$ have deleterious acute cardiovascular effects [14,15], studies suggest that residing in areas with higher long-term average $PM_{2.5}$ levels is associated with an even greater effect on cardiovascular health [16]. Indeed, reductions in annual $PM_{2.5}$ averages have been associated with decreased cardiovascular mortality rates, even when accounting for socioeconomic status [17]. Thus, we utilized average annual $PM_{2.5}$ levels as published by CDC in an effort to study the cardiovascular effects of long-term exposure to $PM_{2.5}$.

### Outcome measures

County data for HF hospitalizations/1000 Medicare beneficiaries (2012–2014) was acquired from the CDC [13]. HF mortality/100,000 (2012–2014) was derived from cardiovascular-related death certificates, which has been shown to be accurate in 82% of the cases [18].

### Demographics

We adjusted for demographic variables to ensure that any observed relationships were not solely due to differences in sex, race, or age in the population. We obtained county data on percentage male, percentage white, and median age from the 2010 US Census Bureau [13].

## Socioeconomic factors

Poverty has been established as being significantly associated with worse outcomes in HF [19] as well as higher exposure to pollution [20], therefore we adjusted for county level poverty data to account for confounding due to socioeconomic differences. County data for percentage of people living in poverty (2014) was obtained from the US Census Bureau and Bureau of Labor Statistics, and the threshold for poverty was an annual salary of $11,670 for a one-person household plus $4,060 for each additional person [21]. Furthermore, we adjusted for county metropolitan (metro) status because metro status is related to both the degree of pollution in an area and health outcomes [11,12]. Counties were defined as either large central metro, fringe metro, medium/small metro, or nonmetro by the Office of Management and Budget [22].

## Prevalence of population comorbidities in the county

We adjusted for established risk factors of HF morbidity and mortality. Long-term exposure to $PM_{2.5}$ has been linked to higher coronary heart disease (CHD) incidence [23] which in turn is a risk factor for HF [24]. County level data for CHD Hospitalizations/100,000 Medicare Beneficiaries (2012–2014) was obtained from the CDC and used as a surrogate for prevalence of CHD in a county [13]. In addition, county data for percentage of the population older than 20 years diagnosed with diabetes mellitus (2013) was obtained from the CDC [13].

## Prevalence of healthcare associated behaviors

We adjusted for healthcare associated behaviors that may be associated with HF outcomes and possibly with risk of exposure to $PM_{2.5}$. Lack of adherence to medications has been demonstrated to be associated with worse outcomes in HF [25,26]. Smoking is another behavioral risk factor for higher HF mortality [27]. Data for percentage of population with self-reported smoking (2015 American Community Survey) and antihypertensive medication nonadherence (2014) was obtained from the CDC [13].

## Analyses

Continuous variables were described using median and range, and categorical variables were described as numbers and percentages and then stratified by quartiles of $PM_{2.5}$. We used weighted Pearson's correlation coefficient to assess the association between $PM_{2.5}$ concentration levels, HF hospitalizations and HF mortality.

We utilized multivariate linear regression modelling to assess the associations between $PM_{2.5}$ concentration levels (as a continuous variable) and HF hospitalizations first, and HF Mortality, next. Subsequently we adjusted for demographics and then socioeconomic factors, prevalence of comorbidities, and prevalence of nonadherence to antihypertensive medications in successive models. All models included county population size as analytic weights. Because data on lack of adherence to antihypertensive medications was extracted from a self-reported survey and thus may overestimate the true adherence to medications [28,29], we removed this variable from our fully adjusted model as a sensitivity analysis. As prevalence of self-reported smoking was published for only a limited number of counties, we further adjusted the models for prevalence of self-reported smoking as a sensitivity analysis. This reduced our sample size from 3,099 to 2,674 counties in the HF hospitalization analysis and from 3,100 to 2,673 counties in the HF mortality analysis. As secondary analyses, we also included the $PM_{2.5}$ level as quartiles instead of a continuous variable in multivariate linear regression modelling with county population as the analytic weight using the same modeling approach as described.

We tested for heterogeneity of effect of $PM_{2.5}$ concentration levels on incidence of HF hospitalizations and HF mortality by metro status by fitting an interaction term as well as by subgrouping the analyses by metro status. Metropolitan areas were divided into two categories: metro (including large/central metro, fringe metro and small/medium metro) and nonmetro. Multivariate linear regression analysis was used to assess the associations between $PM_{2.5}$ concentration levels and HF outcomes with adjustment for demographics, then demographics, socioeconomic factors, prevalence of comorbidities, and prevalence of nonadherence to antihypertensive medications, grouped according to metropolitan category (Metro vs Nonmetro).

These analyses were conducted in STATA SE v16.1. Statistical significance was defined as a two-sided p-value below 0.05.

## Results

Among the 3135 US counties utilized in this study sample, the median $PM_{2.5}$ concentration level was 9.4 μg/m$^3$ (range: 3.0 to 19.7 μg/m$^3$). The prevalence of poverty, diagnosis of diabetes, CHD hospitalization/100,000 Medicare beneficiaries, and self-reported smoking increased with increasing $PM_{2.5}$ quartiles (p < 0.05) (Table 1). County $PM_{2.5}$ concentration level showed a positive correlation with HF hospitalization among Medicare beneficiaries (r = 0.41) (p < 0.05) which was significantly stronger than the correlation with the overall county HF mortality (r = 0.08) (p < 0.05) (Fig 1). The correlation of county $PM_{2.5}$ concentration with CHD hospitalization was modest (r = 0.24) (p < 0.05). County $PM_{2.5}$ concentration level also showed weak positive correlations with prevalence of poverty (r = 0.07) (p < 0.05), diagnosis of diabetes (r = 0.09) (p < 0.05), nonadherence to antihypertensive medications (r = 0.04) (p < 0.05), and self-reported smoking (r = 0.07) (p < 0.05) (Table 2).

The median county HF hospitalization was 13.5/1000 Medicare beneficiaries (range: 0.5 to 48.4) and increased with increasing $PM_{2.5}$ quartiles (p < 0.05). In the multivariate linear regression analysis using $PM_{2.5}$ as a continuous variable, the unadjusted model for HF hospitalizations showed that for every 1 μg/m$^3$ increase in $PM_{2.5}$ concentration level, there was an associated increase of 0.92 HF hospitalizations/1,000 Medicare beneficiaries (95% CI 0.85–0.99) (Table 3). In a fully adjusted model, with adjustment for demographics, socioeconomic factors, prevalence of comorbidities, and prevalence of nonadherence to antihypertensive medications, $PM_{2.5}$ concentration levels remained significantly associated with HF hospitalization rates, showing an associated increase of 0.51 HF hospitalizations/1,000 Medicare beneficiaries (95% CI 0.45–0.57) for every 1 μg/m$^3$ increase in $PM_{2.5}$ concentration level. The association was largely unchanged after removing lack of adherence to antihypertensive medications from the fully adjusted model (0.52 more HF hospitalizations/1,000 Medicare beneficiaries for every 1 μg/m$^3$ increase in $PM_{2.5}$ concentration, 95% CI 0.46–0.58). The association was unchanged after further adjustment for county prevalence of self-reported smoking (0.51 more HF hospitalizations/1,000 Medicare beneficiaries for every 1 μg/m$^3$ increase in $PM_{2.5}$ concentration, 95% CI 0.45–0.57). In the analysis utilizing $PM_{2.5}$ levels as quartiles instead of a continuous variable, there were significant reductions in HF hospitalization risk with Quartiles 1 and 2 (5.25 and 1.94 lower HF hospitalizations/1,000 Medicare beneficiaries, respectively) as compared to the reference (Quartile 4).

The median county HF mortality was 189.5 deaths/100,000 (range: 18 to 708.3), which also increased with increasing $PM_{2.5}$ quartiles (p < 0.05). In the multivariate linear regression analysis using $PM_{2.5}$ as a continuous variable, the unadjusted model for HF mortality showed that for every 1 μg/m$^3$ increase in $PM_{2.5}$ concentration level, there was an associated increase of 1.69 HF deaths/100,000 (95% CI 0.96–2.41) (Table 4). In the fully adjusted model, when adjusted for demographics, socioeconomic factors, prevalence of comorbidities, and

**Table 1. Descriptive statistics of county PM$_{2.5}$ concentration levels and covariates.**

| Variable | No. of Counties | Overall Median (Range) n = 3135 | PM2.5 Quartile 1 Median (Range) n = 782 | PM2.5 Quartile 2 Median (Range) n = 836 | PM2.5 Quartile 3 Median (Range) n = 739 | PM2.5 Quartile 4 Median (Range) n = 750 |
|---|---|---|---|---|---|---|
| PM2.5 Concentration (2014), µg/m³ | 3,107 | 9.4 (3.0–19.7) | 6.4 (3.0–7.7) | 8.8 (7.8–9.4) | 9.9 (9.5–10.4) | 11.2 (10.5–19.7) |
| Population (2010–2014) | 3,134 | 25,893 (73–9,974,203) | 15,744 (290–9,974203) | 24,516 (89–2,448,943) | 31,223 (174–2,570,801) | 36,209 (559–4,269,608) |
| Median Age (2010 Census), years | 3,133 | 41.0 (21.5–66.0) | 41.3 (24.3–61.2) | 41.4 (21.5–66.0) | 40.8 (24.2–56.6) | 40.8 (23.1–59.8) |
| Male (2010 Census), % | 3,135 | 49.5 (43.2–72.1) | 49.9 (43.2–72.1) | 49.5 (45.3–65.6) | 49.4 (44.8–60.9) | 49.4 (45.5–63.7) |
| White (2015 Census), % | 3,133 | 84.6 (0.7–100) | 85.1 (3.8–100) | 85.0 (0.7–99.0) | 82.2 (9.5–99.3) | 85.8 (5.3–99.5) |
| Population Living in Poverty (2010–2014), % | 3,131 | 15.8 (3.2–52.2) | 14.4 (3.7–44.6) | 15.7 (4.0–52.2) | 17.5 (3.2–47.4) | 16.4 (3.9–47.0) |
| Age Adjusted Population with Diabetes > 20 years (2013), % | 3,133 | 9.3 (3.8–20.8) | 8.4 (3.8–16.7) | 9.2 (4.2–17.8) | 10.0 (4.7–19.4) | 9.9 (4.6–20.8) |
| Coronary Heart Disease Hospitalizations/100,000 Medicare Beneficiaries (2012–2014) | 3,127 | 14.2 (5.0–41.9) | 12.5 (6.0–41.9) | 14.0 (5.4–38.4) | 15.1 (5.0–35.7) | 15.2 (7.0–35.2) |
| Population with Reported Smoking (2015), % | 2,706 | 20.8 (3.1–51.1) | 19.1 (4.5–47.1) | 20.3 (3.1–45.6) | 21.9 (6.6–49.2) | 21.6 (7.5–51.1) |
| Anti-hypertensive Medication Nonadherence in Part D Medicare Beneficiaries (2014), % | 3,135 | 26.1 (1.0–56.2) | 25.6 (1.0–56.2) | 25.4 (1.0–47.1) | 27.3 (1.0–41.8) | 26.2 (1.0–47.9) |
| Metropolitan Status Large Central Metro Fringe Metro Small/Medium Metro Nonmetro | 3,133 | 3,133 68 (2.2%) 368 (11.8%) 731 (23.3%) 1,966 (62.8%) | 780 18 (2.3%) 54 (6.9%) 128 (16.4%) 580 (74.4%) | 836 13 (1.6%) 113 (13.5%) 213 (25.5%) 497 (59.4%) | 739 16 (2.2%) 88 (11.9%) 182 (24.6%) 453 (61.3%) | 750 20 (2.7%) 110 (14.7%) 200 (26.7%) 420 (56.0%) |
| Metropolitan Counties (Large central, Fringe, Small/Medium) | 1,167 | 1,167 | 200 | 339 | 286 | 330 |
| Nonmetro Counties | 1,966 | 1,966 | 580 | 497 | 453 | 420 |

P-value for linear trend across PM$_{2.5}$ quartiles was < 0.05.

prevalence of nonadherence to antihypertensive medications, PM$_{2.5}$ concentration levels remain significantly associated with HF mortality rates, showing an associated increase of 0.74 HF deaths/100,000 (95% CI 0.17–1.30) for every 1 µg/m³ increase in PM$_{2.5}$ concentration level. The association was largely unchanged after removing lack of adherence to antihypertensive medications from the fully adjusted model (0.79 more HF deaths/100,000 for every 1 µg/m³ increase in PM$_{2.5}$ concentration, 95% CI 0.22–1.35). The association was unchanged after further adjustment for county prevalence of self-reported smoking (0.80 more HF deaths/100,000 for every 1 µg/m³ increase in PM$_{2.5}$ concentration, 95% CI 0.21–1.38). In the analysis utilizing PM$_{2.5}$ levels as quartiles instead of a continuous variable, there was a significant reduction in HF mortality risk in Quartile 1 (10.5 lower HF deaths/100,000) as compared to the reference (Quartile 4).

We tested for statistical interaction between metro status and the association of HF outcomes with PM$_{2.5}$ concentration. The relationship between metro status and the association of

**(A)**

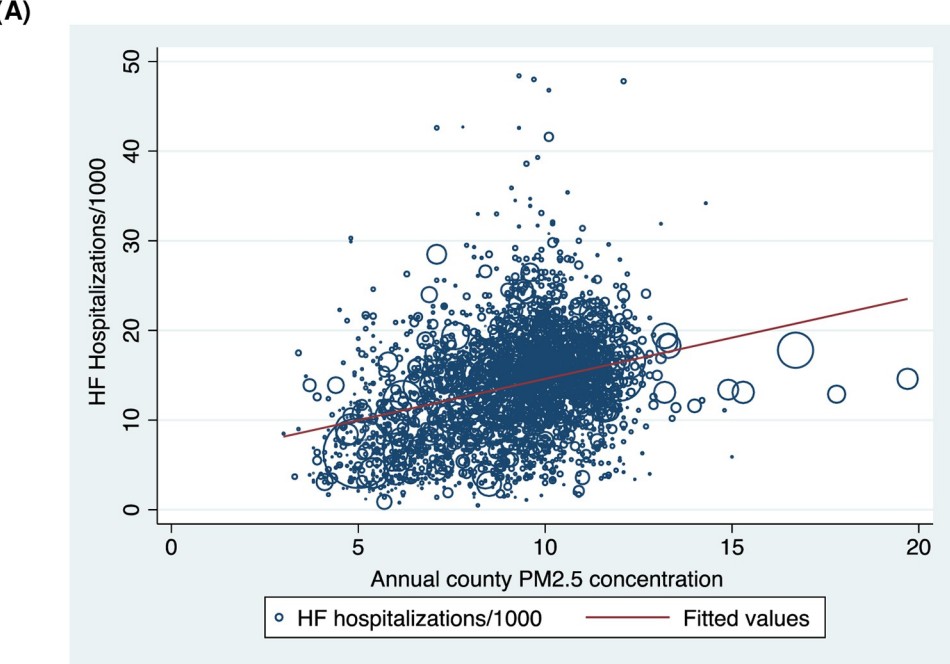

**(B)**

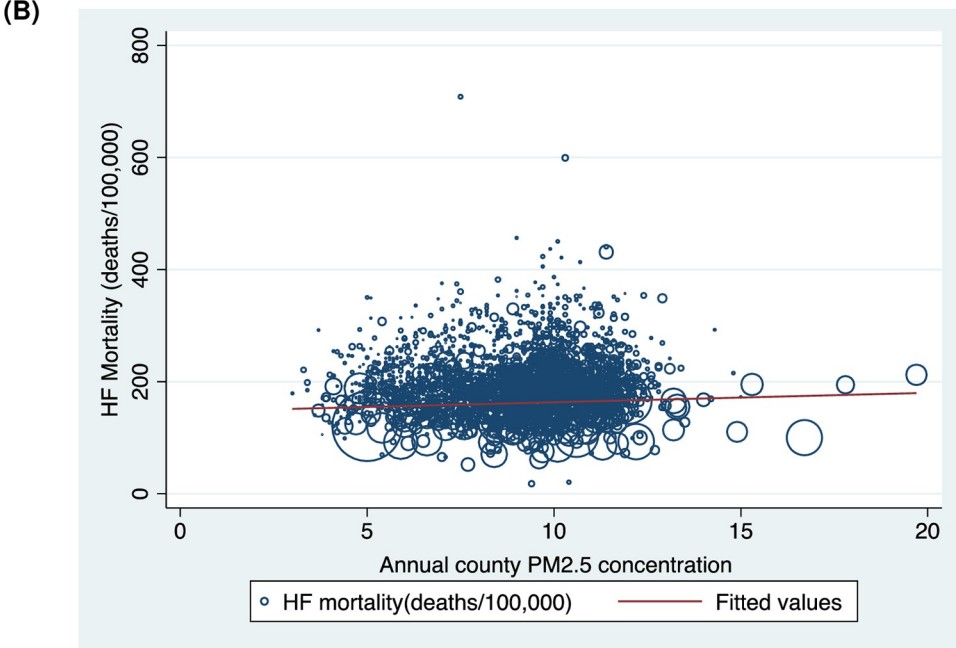

**Fig 1. Correlation of county PM$_{2.5}$ concentration levels and HF outcomes.** Analytic-weighted scatter plots showing correlation of PM$_{2.5}$ concentration levels with (A) heart failure hospitalizations (r = 0.41) (p < 0.05) and (B) heart failure mortality (r = 0.08) (p < 0.05), across 3135 counties of the United States.

HF hospitalization with PM$_{2.5}$ concentration was significant; therefore, we present this association by metro status, comparing metropolitan counties (Large Central, Fringe, and Small/ Medium) to nonmetro counties. In the subgroup analysis by metropolitan status for HF hospitalization, the unadjusted model showed that for every 1 μg/m$^3$ increase in PM$_{2.5}$

**Table 2. Correlation of county PM$_{2.5}$ concentration levels with outcomes and covariates.**

| Variable | Correlation with PM$_{2.5}$ |
|---|---|
| Heart Failure Hospitalizations/1000 Medicare Beneficiaries (2012–2014) | 0.41 |
| Heart Failure Mortality (deaths/100,000) (2012–2014) | 0.08 |
| Median age (2010 census) | -0.04 |
| Percentage male (2010 census) | -0.07 |
| Percentage white (2015 census) | -0.14 |
| Percentage of population living in poverty (2010–2014) | 0.07 |
| Metropolitan status | 0.06 |
| Percentage of population > 20 years diagnosed with diabetes (2013) | 0.09 |
| Coronary Heart Disease Hospitalizations/100,000 Medicare Beneficiaries (2012–2014) | 0.24 |
| Antihypertensive medication nonadherence percentage in Part D Medicare beneficiaries (2014) | 0.04 |
| Percentage population who reported smoking (2015- American Community Survey) | 0.07 |

All P-values < 0.05.

concentration level, there was an associated increase of 0.89 HF hospitalizations/1,000 Medicare beneficiaries (95% CI 0.78–0.99) in the metropolitan counties and 1.11 HF hospitalizations/1,000 Medicare beneficiaries (95% CI 0.97–1.24) in the nonmetro counties (Table 5). In the fully adjusted model, with adjustment for demographics, socioeconomic factors, prevalence of comorbidities, and prevalence of nonadherence to antihypertensive medications, the

**Table 3. HF hospitalizations analytic-weighted multivariate linear regression modeling.**

| | No. of Observations | Overall | Quartile 1 | Quartile 2 | Quartile 3 | Quartile 4 |
|---|---|---|---|---|---|---|
| Heart Failure Hospitalizations/1,000 Medicare Beneficiaries 2012–2014; **Median (Range)** | 3,126 | 13.5 (0.5–48.4) | 9.1 (0.8–42.6) | 13.5 (0.5–48.4) | 15.0 (1.1–48.0) | 15.4 (1.7–47.8) |
| Multivariate Linear Regression Modeling | | | | | | |
| Unadjusted HF Hospitalization and PM$_{2.5}$; **Regression coefficient (95% CI)** | 3,103 | 0.92 (0.85–0.99) | -5.25 (-5.69––4.81) | -1.94 (-2.38––1.50) | -0.23[†] (-0.69–0.22) | Reference |
| Adjusted for Demographics[*]; **Regression coefficient (95% CI)** | 3,102 | 0.89 (0.81–0.96) | -5.15 (-5.59––4.71) | -1.68 (-2.13––1.24) | -0.31[†] (-0.76–0.14) | Reference |
| Adjusted for Demographics, Poverty, Metro Status, Diabetes, CHD Hospitalizations, HTN Meds Nonadherence; **Regression coefficient (95% CI)** | 3,099 | 0.51 (0.45–0.57) | -3.37 (-3.72––3.02) | -0.91 (-1.25––0.56) | -0.17[†] (-0.52–0.18) | Reference |
| Adjusted for Demographics, Poverty, Metro Status, Diabetes, CHD Hospitalizations; **Regression coefficient (95% CI)** | 3,099 | 0.52 (0.46–0.58) | -3.43 (-3.79––3.08) | -0.89 (-1.24––0.54) | -0.20[†] (-0.56–0.15) | Reference |
| Adjusted for Demographics, Poverty, Metro Status, Diabetes, CHD Hospitalizations, HTN Meds Nonadherence + Smoking; **Regression coefficient (95% CI)** | 2,673 | 0.51 (0.45–0.57) | -3.45 (-3.83––3.07) | -0.86 (-1.23––0.48) | -0.17[†] (-0.55–0.20) | Reference |

PM$_{2.5}$ Regression Coefficient represents a per unit increase in 1 μg/m$^3$ above 3.0 μg/m$^3$.

[*] Median Age/County, Percentage Male, Percentage White.

All P-values < 0.05 except for [†].

**Table 4. HF Mortality analytic-weighted multivariate linear regression modeling.**

| | No. of Observations | Overall | Quartile 1 | Quartile 2 | Quartile 3 | Quartile 4 |
|---|---|---|---|---|---|---|
| Heart Failure Mortality (deaths/100,000) 2012–2014; **Median (Range)** | 3,135 | 189.5 (18.0–708.3) | 188 (52.2–708.3) | 184.6 (18.0–456.5) | 196.4 (20.7–599.3) | 192.1 (72.2–440.7) |
| Multivariate Linear Regression Modeling | | | | | | |
| Unadjusted HF Hospitalization and $PM_{2.5}$; **Regression coefficient (95% CI)** | 3,107 | 1.69 (0.96–2.41) | -10.5 (-15.0––6.08) | -0.85[†] (-5.29–3.59) | 3.15[†] (-1.43–7.74) | Reference |
| Adjusted for Demographics[*]; **Regression coefficient (95% CI)** | 3,106 | 2.38 (1.67–3.08) | -12.3 (-16.7––7.99) | -4.64 (-9.00––0.28) | 4.08[†] (-0.39–8.55) | Reference |
| Adjusted for Demographics, Poverty, Metro Status, Diabetes, CHD Hospitalizations, HTN Meds Nonadherence, **Regression coefficient (95% CI)** | 3,100 | 0.74 (0.17–1.30) | -1.99[†] (-5.40–1.42) | -1.61[†] (-4.96–1.74) | 1.03[†] (-2.36–4.42) | Reference |
| Adjusted for Demographics, Poverty, Metro Status, Diabetes, CHD Hospitalizations; **Regression coefficient (95% CI)** | 3,100 | 0.79 (0.22–1.35) | -2.27[†] (-5.67–1.14) | -1.54[†] (-4.89–1.81) | 0.89[†] (-2.50–4.27) | Reference |
| Adjusted for Demographics, Poverty, Metro Status, Diabetes, CHD Hospitalizations, HTN Meds Nonadherence + Smoking; **Regression coefficient (95% CI)** | 2,674 | 0.80 (0.21–1.38) | -3.29[†] (-6.87–0.29) | -1.40[†] (-4.92–2.11) | -0.05[†] (-3.59–3.49) | Reference |

$PM_{2.5}$ Regression Coefficient represents a per unit increase in 1 µg/m$^3$ above 3.0 µg/m$^3$.

[*] Median Age/County, Percentage Male, Percentage White.

All P-values < 0.05 except for [†].

relationship between $PM_{2.5}$ concentration levels and HF hospitalization remained significant, showing an associated increase of 0.51 HF hospitalizations/1,000 Medicare beneficiaries (95% CI 0.42–0.60) in metropolitan counties and 0.56 HF hospitalizations/1,000 Medicare beneficiaries (95% CI 0.46–0.66) in nonmetro counties.

**Table 5. HF hospitalizations subgroup analysis by metropolitan status.**

| Model | No. of Observations | Metro Status | $PM_{2.5}$ Regression Coefficient (95% CI) |
|---|---|---|---|
| Unadjusted HF Hospitalization and $PM_{2.5}$ | 1,153 | Large Central, Fringe, Medium/Small | 0.89 (0.78–0.99) |
| | 1,948 | Nonmetro | 1.11 (0.97–1.24) |
| Adjusted for Demographics[*] | 1,153 | Large Central, Fringe, Medium/Small | 0.86 (0.75–0.96) |
| | 1,947 | Nonmetro | 1.13 (1.00–1.26) |
| Adjusted for Demographics, Poverty, Diabetes, CHD Hospitalization, and HTN Meds Nonadherence | 1,152 | Large Central, Fringe, Medium/Small | 0.51 (0.42–0.60) |
| | 1,947 | Nonmetro | 0.56 (0.46–0.66) |

$PM_{2.5}$ Regression Coefficient represents a per unit increase in 1 µg/m$^3$ above 3.0 µg/m$^3$.

[*] Median Age/County, Percentage Male, Percentage White.

All P-values < 0.05.

## Discussion

To our knowledge, this is the first nationwide study to report the association of long-term exposure to $PM_{2.5}$ with higher HF hospitalization and mortality at the county level and by metro status. We showed a 1 ug/m$^3$ increase in annual county $PM_{2.5}$ levels was associated with an increase of 0.51 HF hospitalizations/1000 Medicare beneficiaries and 0.74 HF deaths/ 100,000 after adjustment for various county level covariates. We also demonstrated that after accounting for metropolitan status of a county, long-term exposure to $PM_{2.5}$ was still associated with higher HF hospitalizations.

Previous US studies have examined the effects of daily exposure to $PM_{2.5}$ on HF hospitalizations; however, these studies have focused only on limited geographic locales and did not study HF mortality [7–10]. Our study builds on these findings, showing nationwide association of $PM_{2.5}$ exposure with both HF hospitalization and mortality accounting for a county's demographics, socioeconomic factors, prevalence of comorbidities, and prevalence of health-care-associated behaviors. There are various proposed biological mechanisms of how $PM_{2.5}$ contributes to HF outcomes, including increased systemic blood pressure and vasoconstriction [30–32], elevated pulmonary and right ventricular diastolic pressures [33], arrythmias [34], and adverse ventricular remodeling and myocardial fibrosis [35]. In addition, air pollution might worsen HF outcomes through its relationship with disease processes that can potentially cause HF, such as CHD and diabetes. Evidence of these relationships are the correlations between diabetes and CHD with $PM_{2.5}$ concentration (Table 2) as well as significant attenuation of the associations after accounting for these potential mechanistic factors. However, further research is necessary to elucidate the various pathophysiologic relationships between $PM_{2.5}$ and HF.

Significant associations between long-term exposure to $PM_{2.5}$ and all-cause mortality in HF patients has also been described among patients in the University of North Carolina Health Care System [36]. However, this study examined only one region of the country, hence unable to encompass a variety of different $PM_{2.5}$ concentrations, influenced by topographical, geographical, and climate differences [37,38]. Another study conducted in the US has shown a significant association between long-term exposure to $PM_{2.5}$ and non-ischemic heart disease mortality, including HF mortality (grouped together with dysrhythmia and cardiac arrest) on a nationwide level [39]. Our study builds on these findings by showing associations of $PM_{2.5}$ with ischemia, such as CHD and diabetes, and added information on intermediate outcomes such as HF hospitalization rates.

It is of note that the association of $PM_{2.5}$ concentration and HF hospitalization was still significant in rural counties. Even with adjustment for various socioeconomic variables, this relationship remained statistically significant. This would indicate that $PM_{2.5}$ levels in the nonmetro counties may be associated with different sources of pollution. For example, the close proximity of Large Central metropolitan areas to roadways and traffic has been shown to be associated with increased cardiovascular disease (CVD) due to increased exposure to combustion-derived particulates, a source of air pollution [40]. However, policies aimed towards reducing outdoor pollution in metro regions are correspondingly more robust. In comparison, rural areas may rely more heavily on burning biomass fuels (e.g., wood, charcoal) for cooking, lighting, and heating, which may contribute to both indoor and outdoor $PM_{2.5}$ [41], leading to the release of particulate matter and other pollutants linked to CVD [42]. This calls for more research regarding the sources of $PM_{2.5}$, indoor versus outdoor $PM_{2.5}$ and their detrimental health effects.

The results of this study call for potential policies aimed towards reducing ambient air pollution in both metro and rural areas. Previous policies implemented by the United States may

be more focused on urban pollutants. In the US, the Clean Air Act of 1970 reduced major industrial pollutants (i.e., particulate matter, sulfur oxides, nitrogen oxides, carbon monoxide, volatile organic compounds, and lead) by 73% over a 25-year span [43]. Reports have linked this reduction in ambient air pollution to fewer acute myocardial infarctions (200,000 fewer cases per year) [44] and improvements in life-expectancy (increase of 7 months) [45] from 1990 to 2020. In July 1990, Hong Kong passed a restriction on sulfur content used by power plants and motor vehicles. Over the following five years, there were decreased annual rates of all-cause mortality (2.1%), respiratory mortality (3.9%), and cardiovascular mortality (2.0%) as well as increased life expectancy (20 days for women, 41 days for men) [46]. More research evaluating the downstream effects of policies aimed at reducing ambient particulate matter pollution on HF outcomes in both urban and rural areas are necessary. The results of this study also set the stage for future research utilizing machine learning, incorporating temporal trends in climate fluctuations and heterogenous levels of industrialization of cities and towns to predict changes in heart failure outcomes of populations and geographic areas.

The strength of our study is that we utilized nationwide data with a large sample size (3135 US counties), allowing for greater generalizability. Secondly, we present the analysis of the effects of chronic exposure to $PM_{2.5}$, which averages out the day-to-day variability due to weather, humidity, and seasonal effects and is less studied compared to short-term exposure. Thirdly, we examined the effects of metropolitan status on the relationship between pollution and heart failure, which has not been previously studied. There are also limitations to consider while reviewing these results. This is a cross sectional analysis utilizing aggregate data instead of individual level data and therefore cannot establish temporal sequence or causality. Although we attempted to account for all possible mediators and confounders by using multivariate modeling, residual confounding factors are likely to exist outside the scope of this study. It must be noted, however, that any association observed at the aggregate level will likely be attenuated due to the scale of the study unit. Additionally, the usage of self-reported data on antihypertensive medication adherence may overestimate the true adherence to medications [28,29]. However, the impact of this limitation is likely minimal since there is no reason to believe that the degree of under-reporting of antihypertensive medication nonadherence will vary across counties of distinct air pollution levels. To confirm, we performed a sensitivity analysis excluding lack of adherence to antihypertensive medications from our fully adjusted models and found no significant change in the associations between $PM_{2.5}$ and heart failure hospitalizations and mortality.

## Acknowledgments

The views expressed in this paper represent the authors and not the Department of the Veterans Affairs.

## Author Contributions

**Conceptualization:** Edward W. Chen, Khansa Ahmad, Sebhat Erqou, Wen-Chih Wu.

**Data curation:** Edward W. Chen, Khansa Ahmad, Wen-Chih Wu.

**Formal analysis:** Edward W. Chen.

**Funding acquisition:** Edward W. Chen.

**Investigation:** Edward W. Chen, Khansa Ahmad, Wen-Chih Wu.

**Methodology:** Edward W. Chen, Khansa Ahmad, Sebhat Erqou, Wen-Chih Wu.

**Project administration:** Wen-Chih Wu.

**Resources:** Wen-Chih Wu.

**Software:** Edward W. Chen.

**Supervision:** Khansa Ahmad, Sebhat Erqou, Wen-Chih Wu.

**Validation:** Khansa Ahmad, Wen-Chih Wu.

**Visualization:** Khansa Ahmad, Wen-Chih Wu.

**Writing – original draft:** Edward W. Chen.

**Writing – review & editing:** Edward W. Chen, Khansa Ahmad, Sebhat Erqou, Wen-Chih Wu.

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
