## [Decision Letter · Decision Letter 0]

25 Oct 2022

PONE-D-22-22463Particulate matter 2.5, metropolitan status, and heart failure outcomes in US counties: a nationwide ecologic analysisPLOS ONE

Dear Dr.Wu,

Thank you for submitting your manuscript to PLOS ONE. After careful consideration, we feel that it has merit but does not fully meet PLOS ONE’s publication criteria as it currently stands. Therefore, we invite you to submit a revised version of the manuscript that addresses the points raised during the review process.

We look forward to receiving your revised manuscript.

Kind regards,

Linglin Xie

Academic Editor

PLOS ONE

Journal Requirements:

Reviewers' comments:

Reviewer's Responses to Questions

**Comments to the Author**

1. Is the manuscript technically sound, and do the data support the conclusions?

Reviewer #1: Partly

Reviewer #2: Yes

2. Has the statistical analysis been performed appropriately and rigorously? 

Reviewer #1: Yes

Reviewer #2: Yes

3. Have the authors made all data underlying the findings in their manuscript fully available?

Reviewer #1: Yes

Reviewer #2: Yes

4. Is the manuscript presented in an intelligible fashion and written in standard English?

Reviewer #1: Yes

Reviewer #2: Yes

5. Review Comments to the Author

Reviewer #1: In the introduction, the authors could provide some statistics and more in-depth background regarding the prevalence of HF. The urgency and significance of this study were not emphasized enough. The number of counties differ for each variable in Table 1, which I presume is due to the fact that the availability of the data differs for each variable. However, to make a more convincing argument, I wish all the variables were derived from a uniform dataset. The authors did mention some limitations with the use of aggregate data, which I see as the main concern in this paper. I also wish that the authors performed more substantial analyses that went beyond the correlation and interaction of multiple variables with HF outcomes (such as if the variables can predict the HF outcome via machine learning).

Reviewer #2: This study is a cross sectional analysis in the US on the association between particulate matter pollutants with 2.5 micrometers or less (PM2.5) and heart failure (HF) hospitalizations and HF mortality. Aggregate data was extracted from the Centers of Disease Control (CDC) and other databases, and different possible confounders were evaluated, such as sex, race, age, socioeconomic factors, metropolitan status and comorbidities like diabetes and coronary heart disease (CHD). Data on lack of adherence to medications regarding HF was extracted form a survey of antihypertensive medication, and perhaps this extrapolation may be inaccurate.

Statistical analysis was adequately described, and multivariate linear regression modelling was used to asses associations between the different variables. The authors found positive correlation between higher levels of PM2.5 and higher rate of HF hospitalizations, as well as higher mortality, even after multivariate analysis. They suggest environmental policies should be developed to reduce urban pollutants.

The limitations of the study are well presented, including the fact it is an ecological study and do not present individual data and do not allow determination of causality.

In conclusion, the study is adequately written and the authors answer the central question.

6. PLOS authors have the option to publish the peer review history of their article (what does this mean?). If published, this will include your full peer review and any attached files.

Reviewer #1: **Yes: **Minsun S. Jeon

Reviewer #2: No

---

## [Author Response · Author response to Decision Letter 0]

23 Nov 2022

REVIEWER 1

We thank the reviewer for their invaluable recommendations. We have addressed the issues and have incorporated edits into the manuscript as detailed below.

1. In the introduction, the authors could provide some statistics and more in-depth background regarding the prevalence of HF. The urgency and significance of this study were not emphasized enough. 

We thank the reviewer for their invaluable contribution and wholeheartedly agree with their assessment. To address this, the following sentences below (and their respective references) regarding heart failure prevalence and cost have been added to the Introduction section to highlight the urgency and significance of this study: 

“Heart failure affects approximately 6.2 million adults in the United States [4] and was noted in 13.4% of all US death certificates in 2018 [4]. The national cost of heart failure was estimated to be $30.7 billion in 2012 [5].”

2. The number of counties differ for each variable in Table 1, which I presume is due to the fact that the availability of the data differs for each variable. However, to make a more convincing argument, I wish all the variables were derived from a uniform dataset. 

We thank the reviewer for highlighting this concern! As stated in the manuscript, all our data was derived from publicly available databases, specifically the CDC Interactive Atlas of Heart Disease and Stroke as well as the US Census Bureau. We agree that it would be ideal if all variables contained data from all US counties. However, we are reassured by the fact that almost all variables contained data from the vast majority of US counties. The only variable that contained significantly less data was “percentage of population who reported smoking” (containing data from only 2,706 out of the possible 3,135 counties). To account for this, we performed a sensitivity analysis where percentage of population who reported smoking was individually added to the regression models, and the results show that regression coefficients remained largely similar (refer to the last rows of Tables 3 and 4).

3. The authors did mention some limitations with the use of aggregate data, which I see as the main concern in this paper. 

We appreciate the reviewer for highlighting this very valid concern! We acknowledge in the discussion section of the manuscript that large aggregate data cannot discern granular biological mechanisms or investigate individual subgroups in detail. However, we would also like to highlight the strengths of such data, such as the nationwide sample that allows for comparative assessments of trends in large social groups as well as geographic regions as results of policies or environmental factors which would otherwise be difficult to ascertain in studies confined to a region, institution, or social group. Our goal with this study is to identify pervasive nationwide associations so that future studies utilizing individual-level data can be performed to better understand the biologic pathways or the interaction between disease processes that might be underlying these populational trends.

4. I also wish that the authors performed more substantial analyses that went beyond the correlation and interaction of multiple variables with HF outcomes (such as if the variables can predict the HF outcome via machine learning).

We greatly appreciate the reviewer’s suggestion, and we agree that machine learning can be a very fruitful and productive method of analysis for prediction of outcomes. However, this study will serve as the basis for future analyses that use machine learning which incorporates temporal trends in climate fluctuations and heterogeneous levels of industrialization of cities and towns to predict changes in heart failure outcomes of populations and geographic areas. To highlight this potential future direction, the following statement was added to the discussion section: “The results of this study also set the stage for future research utilizing machine learning, incorporating temporal trends in climate fluctuations and heterogenous levels of industrialization of cities and towns to predict changes in heart failure outcomes of populations and geographic areas.”

REVIEWER 2

We are appreciative of the reviewer’s kind remarks and constructive insights. We have addressed any issues and have incorporated edits into the manuscript as detailed below.

1. This study is a cross sectional analysis in the US on the association between particulate matter pollutants with 2.5 micrometers or less (PM2.5) and heart failure (HF) hospitalizations and HF mortality. Aggregate data was extracted from the Centers of Disease Control (CDC) and other databases, and different possible confounders were evaluated, such as sex, race, age, socioeconomic factors, metropolitan status and comorbidities like diabetes and coronary heart disease (CHD). Data on lack of adherence to medications regarding HF was extracted form a survey of antihypertensive medication, and perhaps this extrapolation may be inaccurate. Statistical analysis was adequately described, and multivariate linear regression modelling was used to assess associations between the different variables. The authors found positive correlation between higher levels of PM2.5 and higher rate of HF hospitalizations, as well as higher mortality, even after multivariate analysis. They suggest environmental policies should be developed to reduce urban pollutants. The limitations of the study are well presented, including the fact it is an ecological study and do not present individual data and do not allow determination of causality. In conclusion, the study is adequately written and the authors answer the central question.

We thank the reviewer for highlighting potential limitations of self-reported data on anti-hypertensive medication adherence as they often overestimate the true adherence to medications [1,2]. Since there is no reason to believe that the degree of under-reporting of anti-hypertensive medication non-adherence will vary across counties of distinct air pollution levels, the impact of such a variable is likely minimal. To confirm this point, we performed a sensitivity analysis excluding the lack of adherence to the anti-hypertensive medication variable and found no significant change in the association between PM2.5 levels and heart failure hospitalizations or mortality. 

To reflect these findings, the following additions were made to the manuscript:

- “Because data on lack of adherence to antihypertensive medications was extracted from a self-reported survey and thus may overestimate the true adherence to medications [28,29], we removed this variable from our fully adjusted model as a sensitivity analysis.” was added to the Methods section.

- “The association was largely unchanged after removing lack of adherence to antihypertensive medications from the fully adjusted model (0.52 more HF hospitalizations/1,000 Medicare beneficiaries for every 1 �g/m3 increase in PM2.5 concentration, 95% CI 0.46 – 0.58).” was added to the results section corresponding to Table 3.

- “The association was largely unchanged after removing lack of adherence to antihypertensive medications from the fully adjusted model (0.79 more HF deaths/100,000 for every 1 �g/m3 increase in PM2.5 concentration, 95% CI 0.22 – 1.35).” was added to the results section corresponding to Table 4.

- An extra row was added to Tables 3 and 4 reflecting this sensitivity analysis.

- “Additionally, the usage of self-reported data on antihypertensive medication adherence may overestimate the true adherence to medications [28,29]. However, the impact of this limitation is likely minimal since there is no reason to believe that the degree of under-reporting of antihypertensive medication nonadherence will vary across counties of distinct air pollution levels. To confirm, we performed a sensitivity analysis excluding lack of adherence to antihypertensive medications from our fully adjusted models and found no significant change in the associations between PM2.5 and heart failure hospitalizations and mortality." was added to the limitations paragraph of the discussion section.

References

1. Stirratt MJ, Dunbar-Jacob J, Crane HM, Simoni JM, Czajkowski S, Hilliard ME, Aikens JE, Hunter CM, Velligan DI, Huntley K, et al. Self-report measures of medication adherence behavior: recommendations on optimal use. Transl Behav Med. 2015;5:470-482. doi: 10.1007/s13142-015-0315-2

2. Berg KM, Arnsten JH. Practical and conceptual challenges in measuring antiretroviral adherence. J Acquir Immune Defic Syndr. 2006;43 Suppl 1:S79-87. doi: 10.1097/01.qai.0000248337.97814.66

---

## [Editor Report · Decision Letter 1]

14 Dec 2022

Particulate matter 2.5, metropolitan status, and heart failure outcomes in US counties: a nationwide ecologic analysis

PONE-D-22-22463R1

Dear Dr. Wu,

We’re pleased to inform you that your manuscript has been judged scientifically suitable for publication and will be formally accepted for publication once it meets all outstanding technical requirements.

Kind regards,

Linglin Xie

Academic Editor

PLOS ONE
---

## [Editor Report · Acceptance letter]

22 Dec 2022

PONE-D-22-22463R1 

Particulate matter 2.5, metropolitan status, and heart failure outcomes in US counties: a nationwide ecologic analysis 

Dear Dr. Wu:

I'm pleased to inform you that your manuscript has been deemed suitable for publication in PLOS ONE. Congratulations! Your manuscript is now with our production department. 

Kind regards, 

on behalf of

Dr. Linglin Xie 

Academic Editor

PLOS ONE